# A Rescue Use of ECPELLA for Sepsis-Induced Cardiogenic Shock Followed by Mitral Valve Replacement

**DOI:** 10.3390/medicina58060698

**Published:** 2022-05-25

**Authors:** Makiko Nakamura, Teruhiko Imamura, Akira Oshima, Mitsuo Sobajima, Shigeki Yokoyama, Toshio Doi, Kazuaki Fukahara, Koichiro Kinugawa

**Affiliations:** 1The Second Department of Internal Medicine, University of Toyama, Toyama 930-0194, Japan; nakamura@med.u-toyama.ac.jp (M.N.); brother0917jp@gmail.com (A.O.); soba1126@yahoo.co.jp (M.S.); kinugawa@med.u-toyama.ac.jp (K.K.); 2Department of Cardiovascular Surgery, University of Toyama, Toyama 930-0194, Japan; yoko1@med.u-toyama.ac.jp (S.Y.); tdoi@med.u-toyama.ac.jp (T.D.); fuka@med.u-toyama.ac.jp (K.F.)

**Keywords:** mechanical circulatory support, septic shock, left ventricular assist device, mitral chordal rupture, septic cardiomyopathy

## Abstract

The use of veno-arterial extracorporeal membrane oxygenation (ECMO) in patients with sepsis-induced cardiogenic shock has been reported, but the clinical implication of the Impella percutaneous axial-flow left ventricular assist device for such patients remains unknown. We had a 37-year-old man with septic shock and severely reduced cardiac function. Veno-arterial ECMO and concomitant Impella CP support ameliorated his end-organ dysfunction and achieved cardiac recovery, whereas severe mitral valve regurgitation due to chordal rupture developed later. Mitral valve replacement concomitant with ECMO removal as well as an Impella upgrade successfully treated the patient. ECMO and Impella support might be an effective therapeutic strategy for the bridge to recovery in patients with sepsis-induced cardiogenic shock; however, paying attention to mitral chordal rupture is highly encouraged.

## 1. Introduction

Approximately 15% of patients with sepsis develop septic shock, which is defined as persistent hypotension and elevated lactate levels refractory to adequate fluid resuscitation and requiring vasopressors. Its in-hospital mortality exceeds 40% [1]. Transient myocardial dysfunction is its common comorbidity. It increases the mortality up to 70–90%, and its incidence varies from 18% to 40% [2]. The pathology is defined as sepsis-induced cardiogenic shock.

A retrospective cohort study confirmed the survival improvement by using extracorporeal membrane oxygenation (ECMO) in the cohort with sepsis-induced cardiogenic shock [3], whereas a recent meta-analysis clarified that ECMO had rather worse outcomes in patients with septic shock without severely decreased cardiac function [4]. ECMO is considered a suitable treatment option in selected patients with severe sepsis-induced cardiogenic shock [2], although the optimal patient selection procedure requires further investigation.

The clinical implication of percutaneous axial-flow left ventricular assist device Impella in patients with sepsis-induced cardiogenic shock has not yet been investigated. ECMO alone would be insufficient, particularly for those with severe hepatorenal dysfunction and congestive heart failure with severely impaired left ventricular function. Given the accumulating evidence of Impella with/without ECMO in patients with cardiogenic shock, Impella-incorporated mechanical circulatory support therapy might also have a promising outcome for those with sepsis-induced cardiogenic shock.

## 2. Case Report

### 2.1. Before Referral

A 37-year-old man without any medical history visited the former out-patient clinic complaining of high fever, swelling of the neck on the right side, and appetite loss. He had no symptoms of cough and sore throat, as well as pharyngeal exudate and sputum. He was diagnosed with cervical lymphadenitis, and oral antibiotics and non-steroidal anti-inflammatory drugs were prescribed. However, his symptoms worsened, and he was readmitted to the same hospital three days following the clinic visit. Despite intravenous antibiotics and corticosteroid administration, his systolic blood pressure decreased to 60 mmHg, which was refractory to the administration of massive fluid and norepinephrine. The next morning, laboratory data showed an elevation of serum creatinine (3.7 mg/dL) and total bilirubin (5.4 mg/dL). A transthoracic echocardiogram showed a reduced left ventricular ejection fraction of 20% with diffuse severe hypokinesis. Intravenous dobutamine infusion was initiated. He was transferred to our hospital for further treatment.

### 2.2. On Admission

On arrival, his blood pressure was 83/63 mmHg with a heart rate of 128 bpm under intravenous dobutamine and norepinephrine infusion. His body temperature was 38.9 degrees, and oxygen saturation was 96%. Cardiac Troponin I was 494.8 pg/mL (>26.2) with normal serum creatine kinase of 102 ng/mL. Serum creatinine (4.26 mg/dL), total bilirubin (4.8 mg/dL), white blood cell (13,400/mm^3^), C-reactive protein (22.64 mg/dL), and procalcitonin (6.72 U/mL) (>0.05) were markedly elevated. The levels of plasma B-type natriuretic peptide (494.8 pg/mL) and serum N-terminal-pro-B-type natriuretic peptide (30,821 pg/mL) were also elevated. The lactate was 2.5 mmol/L. A polymerase chain reaction test for SARS-CoV-2 was negative.

A chest X-ray showed cardiomegaly with mild pulmonary congestion (Figure 1A). AN electrocardiogram displayed no ST-segment elevation (Figure 1B). An urgent coronary angiogram denied significant stenosis. Right ventricular endomyocardial biopsy showed monocyte infiltrations, consisting of scattered CD8-positive T lymphocytes and CD68-positive macrophages. The right atrial pressure was 15 mmHg, pulmonary artery pressure was 34/26 mmHg, pulmonary artery wedge pressure was 26 mmHg, and the left ventricular end-diastolic pressure was 31 mmHg. The cardiac index was 2.82 L/min/m^2^, the right ventricular stroke work index was 4.4 g/m, and the mixed venous oxygen saturation was 59.4%. He was diagnosed with septic shock and congestive heart failure with severe biventricular dysfunction.

### 2.3. Veno-Arterial ECMO and Impella Support

Veno-arterial ECMO was inserted via femoral vessels to improve his end-organ dysfunction (Figure 2). Impella CP was also inserted via the femoral artery concomitantly to increase his cardiac output and systemic perfusion as well as unload his left ventricle. Anuria persisted, and continuous hemodiafiltration was initiated. He had tooth decay, and we suspected of deep neck infection. Meropenem, cefazoline, and metronidazole were administered.

His serum creatinine and total bilirubin levels gradually improved with their maximum levels of 7.3 mg/dL and 22.8 mg/dL, respectively. The left ventricular ejection fraction improved up to 40%, as well as the pulmonary artery pulsatility index to 1.2 on day 4. However, mitral valve regurgitation increased from trivial to moderate (Figure 2).

### 2.4. ECMO Weaning and Mitral Valve Replacement

On day 5, mitral valve regurgitation progressed to severe and transesophageal echocardiography demonstrated the prolapse of the posterior leaflet and a chordal rupture of posterior commissure (Figure 3A,B), accompanied by the elevation of lactate dehydrogenase up to 1533 IU/L. The bacterial cultures remained sterile, and the C-reactive protein tended to decline.

Emergent mitral valve replacement concomitant with the removal of peripheral veno-arterial ECMO and Impella CP was performed on day 6. He could be weaned from cardiopulmonary bypass with Impella 5.0 support inserted via the axillary artery without the requirement of central ECMO. Mechanical ventilation was weaned off on day 10.

### 2.5. Impella Weaning with Cardiac Recovery

Renal replacement therapy was weaned off on day 14, and Impella 5.0 was weaned off on day 17, respectively. Five tooth cavities that were the potential source of infection were extracted. Meropenem was de-escalated to ceftriaxone, and these antibiotics were administered for six weeks in total. He was administered enalapril, carvedilol, eplerenone, and warfarin.

He was discharged on foot on day 42 with a serum creatinine of 0.95 mg/dL, total bilirubin of 0.9 mg/dL, a C-reactive protein of 0.04 mg/dL, and a plasma B-type natriuretic peptide of 20.9 pg/mL. His left ventricular ejection fraction was 67% at one month following the index discharge.

## 3. Discussion

### 3.1. Sepsis-Induced Cardiogenic Shock

Blood culture tests were not performed beforehand, whereas the Sepsis-related Organ Failure Assessment score was five points upon admission [1]. Given his persistent hypotension and elevated lactate level, we diagnosed him with septic shock.

Transient myocardial dysfunction during septic shock is not rare, showing a mortality rate of 70–90% [2]. Several risk factors are reported: male sex, younger age, elevated lactate levels, and a history of heart failure [5]. Sepsis-induced bacterial toxins, cytokines (tumor necrosis factor, interleukin-1, and interleukin-6), nitric oxide, and cardiac mitochondrial dysfunction seem to be associated with the deterioration of cardiac contractility [2].

Interleukin-6 measured on day 5 was highly elevated up to 160 pg/mL (>8.0) in our patient. Macrophage-predominant monocyte infiltration was observed in his myocardium. Although the definite diagnostic criteria remain unestablished, we defined him as having sepsis-induced cardiogenic shock.

### 3.2. Mechanical Circulatory Support in Sepsis-Induced Cardiogenic Shock

A retrospective cohort study for patients with sepsis-induced cardiogenic shock showed a greater improvement in survival for patients treated with veno-arterial ECMO [3]. A case series of 52 patients treated with ECMO reported that no patients aged above 60 years survived. In another study, those with a history of cardiopulmonary resuscitation prior to ECMO support had worse outcomes [6]. The other cases of 151 patients reported that a door-to-ECMO time of >96 h was associated with worse outcomes [7]. Optimal patient selection is required for therapy with ECMO alone, and further optimal mechanical circulatory support therapies would be required for better outcomes.

Our patient was young, without prior cardiopulmonary resuscitation, and had advanced biventricular dysfunction with severe hepatorenal dysfunction on admission. ECMO alone would increase the afterload on his left ventricle and delay the recovery of the left ventricular ejection fraction as well as worsen his pulmonary congestion, whereas ECMO unloads the right ventricle and increases systemic perfusion. ECMO, concomitant with Impella support, provided a sufficiently high cardiac output of 7.5 L/min in total, ameliorated end-organ dysfunction, unloaded the right and left ventricle, and facilitated the weaning off of ECMO [6].

### 3.3. Mitral Chordal Rupture and Mitral Valve Replacement

Iatrogenic acute mitral regurgitation, as a complication of Impella CP and 5.0, following acute myocardial infarction has been reported [8,9]. Mitral chordal rupture induced by Impella 5.0 in a patient with fulminant myocarditis, probably due to inflammatory vulnerability of valve leaflet and chordae, was also reported [10].

The Mitral chordal rupture that our patient encountered would have also been triggered by a similar mechanism. We selected a mechanical valve, which has no protrusions under the mitral valve and less possibility to interface with Impella 5.0. Specific caution should be paid to avoid interfacing between the intra-cardiac devices, including the Impella, and the cardiac structure, including the mitral valve, to avoid such a complication.

## 4. Conclusions

Support with Impella and veno-arterial ECMO might be a viable rescue strategy in sepsis-induced cardiogenic shock, as long as we pay caution to the complication of mitral chordal rupture.

## Figures and Tables

**Figure 1 medicina-58-00698-f001:**
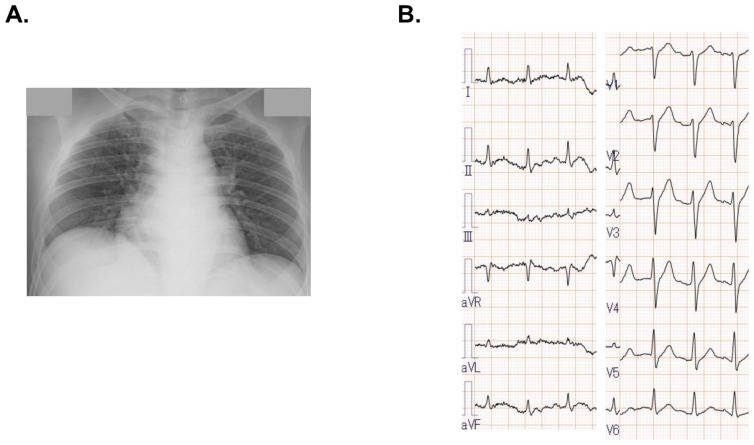
Chest X-ray on admission showed cardiomegaly and mild pulmonary congestion (**A**). Electrocardiogram showed sinus rhythm with no ST-segment elevation (**B**).

**Figure 2 medicina-58-00698-f002:**
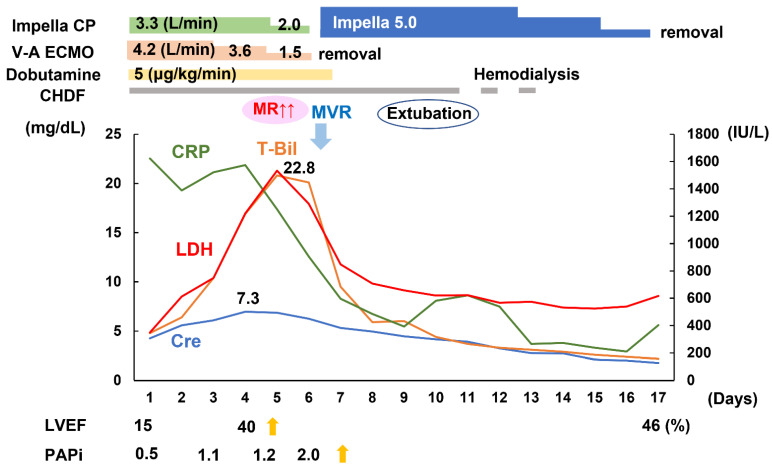
Clinical course after initiation of veno-arterial ECMO and Impella supports. CRP, C-reactive protein; LDH, Lactate dehydrogenase; T-Bil, Total bilirubin; Cre, Creatinine, LVEF, left ventricular ejection fraction; PAPi; Pulmonary artery pulsatility index; V-A ECMO, Veno-arterial extracorporeal membrane oxygenation; CHDF, continuous hemodiafiltration; MR; Mitral valve regurgitation; MVR; Mitral valve replacement.

**Figure 3 medicina-58-00698-f003:**
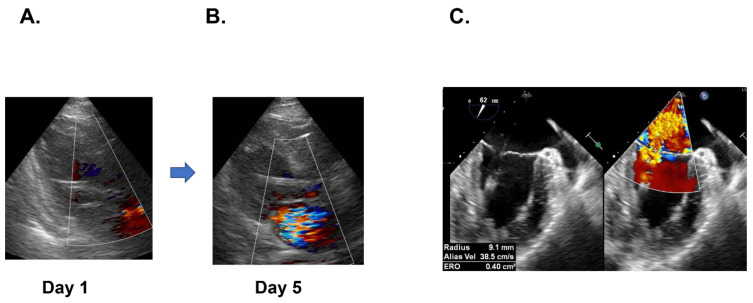
Mitral valve regurgitation was trivial on day 1 (**A**), however, it worsened to severe on day 5 on transthoracic echocardiography (**B**). Transesophageal echocardiography revealed the severe mitral valve regurgitation with prolapse of posterior leaflet and chordal rupture of posterior commissure (**C**).

## Data Availability

Data are contained within the article.

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
