# Peer review of "A Rescue Use of ECPELLA for Sepsis-Induced Cardiogenic Shock Followed by Mitral Valve Replacement"

_medicina, 2022, doi:10.3390/medicina58060698_

Round 1

Reviewer 1 Report

The authors present an interesting case report regarding the use of ECMO and Impella support as an effective therapeutic strategy for bridge to recovery in patients with sepsis-induced cardiogenic shock, with the risk of developing severe mitral valve regurgitation due to chordal rupture. I consider that this is an interesting case report, with clear and legible figures and interesting findings that can have important implications for clinical practice. I consider that the article is valuable and sound and can be published after some minor revisions.

  1. Have additional investigations been performed for a possible cause of recent upper respiratory tract infection (example: pharyngeal exudate, sputum examination)?
  2. What treatment did the patient receive upon discharge from the hospital?

Author Response

General comment

The authors present an interesting case report regarding the use of ECMO and Impella support as an effective therapeutic strategy for bridge to recovery in patients with sepsis-induced cardiogenic shock, with the risk of developing severe mitral valve regurgitation due to chordal rupture. I consider that this is an interesting case report, with clear and legible figures and interesting findings that can have important implications for clinical practice. I consider that the article is valuable and sound and can be published after some minor revisions.

Response

We sincerely express our great appreciation for the reviewer’s recommendations to our manuscript. According to the reviewer’s comment, we revised our manuscript. We further corrected typographs and grammatical errors. Please read through our revised manuscript.

Comment 1

Have additional investigations been performed for a possible cause of recent upper respiratory tract infection (example: pharyngeal exudate, sputum examination)?

Response 1

Thank you for the question. The patients did not have symptoms suggesting recent upper respiratory tract infection, such as the cough and sore throat as well as pharyngeal exudate and sputum. The sputum culture on admission could not be obtained because he did not have sputum at that time. He had high fever and swelling of the right neck, and was diagnosed as cervical lymphadenitis, that is deep neck infection, which refers to an infection or abscess (collection of pus) located deep in the neck (deep cervical space) under the skin near blood vessels, nerves, and muscles. He had severe tooth decay, which was regarded as possible cause of infection. We suspected of deep neck infection caused by tooth decay. We revise the manuscript as follows.

Before 1

Before referral:

A 37-year-old man without any medical history visited the former out-patient clinic complaining of high fever, swelling of the right neck, and appetite loss. He was diagnosed with cervical lymphadenitis, and oral antibiotics and non-steroidal anti-inflammatory drugs were prescribed.

Veno-arterial ECMO and Impella support:

He had tooth decay, and meropenem, cefazoline, and metronidazole were administered.

After 1

Before referral:

A 37-year-old man without any medical history visited the former out-patient clinic complaining of high fever, swelling of the right neck, and appetite loss. He had no symptoms of cough and sore throat as well as pharyngeal exudate and sputum. He was diagnosed with cervical lymphadenitis, and oral antibiotics and non-steroidal anti-inflammatory drugs were prescribed.

Veno-arterial ECMO and Impella support:

He had tooth decay, and we suspected of deep neck infection. Meropenem, cefazoline, and metronidazole were administered.

Comment 2

What treatment did the patient receive upon discharge from the hospital?

Response 2

Thank you for the important question. He was initiated enalapril, carvedilol, eplerenone, and warfarin as anti-heart failure drugs as well as preventing thrombosis of replaced mechanical valve. We added the following sentences in the text.

Before 2

Impella weaning with cardiac recovery:

Meropenem was de-escalated to ceftriaxone and these antibiotics were administered for six weeks in total. He was discharged on foot on day 42 with serum creatinine of 0.95 mg/dL, total bilirubin of 0.9 mg/dL, C-reactive protein of 0.04 mg/dL, and plasma B-type natriuretic peptide of 20.9 pg/mL.

After 2 

Impella weaning with cardiac recovery:

Meropenem was de-escalated to ceftriaxone and these antibiotics were administered for six weeks in total. He was initiated enalapril, carvedilol, eplerenone, and warfarin. He was discharged on foot on day 42 with serum creatinine of 0.95 mg/dL, total bilirubin of 0.9 mg/dL, C-reactive protein of 0.04 mg/dL, and plasma B-type natriuretic peptide of 20.9 pg/mL.

Reviewer 2 Report

A-V ECMO is a commonly accepted and increasingly used method of treating shock, regardless of the cause. The correctly selected moment of introducing the support shown by the authors is crucial for the effectiveness of the therapy.

Iatrogenic damage is the most undesirable event in medicine. The paper shows clearly the risk of using more and more advanced methods that create the possibility of damage to heart structures.

It is not said what was the access route to implant ECMO and Impella.

There is no discussion about the possible change of the assist devices systems during the operation of replacement the mitral valve. In many centers, the method of choice ECMO connection during open heart surgery is central ECMO. Such ECMO in combination with a left vent gives support closest to physiological and minimizes the risk of heart damage. 

Author Response

General comment

A-V ECMO is a commonly accepted and increasingly used method of treating shock, regardless of the cause. The correctly selected moment of introducing the support shown by the authors is crucial for the effectiveness of the therapy. Iatrogenic damage is the most undesirable event in medicine. The paper shows clearly the risk of using more and more advanced methods that create the possibility of damage to heart structures.

Response

We sincerely express our great appreciation for the reviewer’s comments. We revised our manuscript according to the reviewer’s comment. Please read through our revised manuscript that we believe has improved considerably thanks to the suggestion.

Comment 1

It is not said what was the access route to implant ECMO and Impella.

Response 1

We sincerely express our great appreciation for the reviewer’s comment. We added the actual access sites of veno-arterial ECMO and Impella CP to the manuscript.

Before 1

Veno-arterial ECMO and Impella support:

Veno-arterial ECMO support was initiated to improve his end-organ dysfunction (Figure 2). Impella CP was inserted concomitantly to increase his cardiac output and systemic perfusion as well as unload his left ventricle.

After 1 

Veno-arterial ECMO and Impella support:

Veno-arterial ECMO was inserted via femoral vessels to improve his end-organ dysfunction (Figure 2). Impella CP was also inserted via femoral artery concomitantly to increase his cardiac output and systemic perfusion as well as unload his left ventricle.

Comment 2

There is no discussion about the possible change of the assist devices systems during the operation of replacement the mitral valve. In many centers, the method of choice ECMO connection during open heart surgery is central ECMO. Such ECMO in combination with a left vent gives support closest to physiological and minimizes the risk of heart damage. 

Response 2

We sincerely express our great appreciation for the reviewer’s comments. As the reviewer’s comment, central ECMO with LV vent is the most selected circulatory support system for postcardiotomy cardiogenic shock. We had also planned him to initiate central ECMO support in addition to Impella 5.0, in case of difficulty in weaning from cardiopulmonary bypass, however, his right ventricular function was improved, and he could be weaned from cardiopulmonary bypass with Impella 5.0 support alone. We added the above description to the manuscript.

Before 2

ECMO weaning and mitral valve replacement:

Emergent mitral valve replacement concomitant with Impella upgrade to 5.0 as well as veno-arterial ECMO removal was performed on day 6.

After 2 

ECMO weaning and mitral valve replacement:

Emergent mitral valve replacement concomitant with removal of peripheral veno-arterial ECMO and Impella CP was performed on day 6. He could be weaned from cardiopulmonary bypass with Impella 5.0 support inserted via axillary artery without the requirement of central ECMO.
